# Crack Detection and Localisation in Steel-Fibre-Reinforced Self-Compacting Concrete Using Triaxial Accelerometers

**DOI:** 10.3390/s21062044

**Published:** 2021-03-14

**Authors:** Jeffri Ramli, James Coulson, James Martin, Brabha Nagaratnam, Keerthan Poologanathan, Wai Ming Cheung

**Affiliations:** Department of Mechanical and Construction Engineering, Northumbria University, Newcastle upon Tyne NE1 8ST, UK; jamescoulson97@gmail.com (J.C.); james.e.martin@northumbria.ac.uk (J.M.); brabha.nagaratnam@northumbria.ac.uk (B.N.); keerthan.poologanathan@northumbria.ac.uk (K.P.); wai.m.cheung@northumbria.ac.uk (W.M.C.)

**Keywords:** triaxial accelerometers, bending, crack detection, self-compacting concrete, steel fibres

## Abstract

Cracking in concrete structures can significantly affect their structural integrity and eventually lead to catastrophic failure if undetected. Recent advances in sensor technology for structural health monitoring techniques have led to the development of new and improved sensors for real-time detection and monitoring of cracks in various applications, from laboratory tests to large structures. In this study, triaxial accelerometers have been employed to detect and locate micro- and macrocrack formation in plain self-compacting concrete (SCC) and steel-fibre-reinforced SCC (SFRSCC) beams under three-point bending. Experiments were carried out with triaxial accelerometers mounted on the surface of the beams. The experimental results revealed that triaxial accelerometers could be used to identify the locations of cracks and provide a greater quantity of useful data for more accurate measurement and interpretation. The study sheds light on the structural monitoring capability of triaxial acceleration measurements for SFRSCC structural elements that can act as an early warning system for structural failure.

## 1. Introduction

Self-compacting concrete (SCC) is an innovative type of concrete that has the ability to flow under its own weight without segregation or bleeding, and to fill all areas and corners of the formwork [1,2]. Therefore, it is a suitable construction material for structures with complex geometry or highly congested reinforcement [3,4]. In addition, SCC provides excellent deformability and adequate viscosity due to high paste volume and cement content, a low water-to-cement ratio (w/c), small maximum aggregate, a finer-combined aggregate, and the use of a high-range superplasticiser (SP) [2,5,6].

The use of fibres in concrete has grown significantly due to its technical and economic advantages in the construction industry [7]. Fibres made of steel, glass, carbon, and synthetic fibres of various shapes and surfaces and their effects on the fresh and mechanical properties of SCC have been studied by various researchers [8,9,10,11,12]. The addition of fibres in SCC is said to reduce its workability [13,14,15]—that is, its passing and filling ability, as well as its segregation resistance. However, fibres in SCC offer beneficial improvements in terms of ductility, toughness, and energy absorption capacity [16,17,18]. Fibres also control the crack propagation in concrete, contributing to better post-cracking behaviour [16,19]. In some specific applications, fibres can partially or completely replace conventional reinforcement [20]. Hence, SFRSCC is a very promising construction material with a high potential of application that benefits from the potentials of both SCC and randomly dispersed steel fibres [6].

Over the past decade, extensive research has been conducted on SFRSCC in terms of its workability, mechanical properties, and structural performance with increasing usage in the construction industry [14,21,22,23,24,25,26,27]. Its usage expands to both non-structural and structural applications, such as industrial floors, roads and pavements, sprayed concrete, overlays, composite slabs on steel decking, tunnel lining segments, and precast elements [28]. Over time, however, many of these concrete structures may be exposed to extreme structural loads, climate, and weather conditions. As a result of these conditions, concrete structures could suffer from severe damage or even structural failure due to stress, cracking, and deformation [29].

The formation of macrocracks in concrete is a multiscale process that initiates under tensile stress with the formation of microcracks [30]. Fracture could occur and macrocracks become unstable, which can cause brittle failure [30]. This brittle failure can be prevented by incorporating various fibres in concrete to help control crack formation and propagation [31]. According to Soulioti [32], there are three stages of the fracture process in fibre-reinforced concrete (FRC) under bending: (i) stage 1—microcracking stage; (ii) stage 2—microcracks grow and form macrocracks; and (iii) stage 3—rapid propagation of macrocracks that eventually cause failure. Therefore, it is essential to detect these cracks and their propagation as early as possible prior to structural failure.

The X-ray computed tomography (CT) method, the ultrasonic rebound method, resistance strain gauges, and visual images are some of the conventional methods employed to monitor cracks in concrete [33]. However, these techniques are dependent on manual assistance and prone to hysteresis errors, thus cannot be rely upon for accurate and reliable monitoring [33]. Acoustic emission (AE) testing is a non-destructive testing technique that has been widely used for real-time monitoring and damage assessment of concrete structures [32,34,35,36,37,38]. AE refers to the propagation of transient elastic waves due to the rapid release of energy from a localised source within a material [39]. These elastic waves, also known as AE signals, are detected by one or more sensors placed on the surface of the material. These sensors are usually transducers that convert incident elastic waves into electric signals. A suitable analysis of the AE waveform parameters, namely the peak amplitude, duration time, and frequency, provides various information such as the pattern of cracks in concrete, the amount of energy released, the modes of fracture, and the critical conditions prior to final failure [40]. In addition, multiple AE sensors can be used to accurately identify the location of damage in the concrete based on the differences in the arrival times of the AE signals [41]. A number of earlier researchers have focused on the characterisation of the fracture mode using AE methods in laboratory conditions in bending of plain concrete [40,42] and concrete reinforced with metal bars [43,44], with steel fibres [45,46], as well as with synthetic fibres [47]. Most of these studies are focused on the use of multiple single-axial AE sensors at various locations of the concrete specimens. To monitor the cracks and accurately analyse the three-dimensional (3D) source locations, each of the single-axial sensors needs to be placed in the X, Y, and Z axis on the surface of the concrete specimens [48]. However, this can be time-consuming when multiple single axial sensors are required to be mounted on large concrete structures, and a vast amount of data would be generated, resulting in increases in the storage cost for the structural health monitoring.

Previous research has tended to look at single-axis accelerometers with a single application or multiple single-axis accelerometers fitted to multiple surfaces [34,49]. A single axis is not able to determine the location of the cracking and tends to use ‘hit counts’ [32,50]. These ‘hits’ are a cumulative count of how many times a specific frequency is identified by the accelerometer and does not look further into the depth and the duration of the single crack, or the behaviour of the concrete in specific locations. It is a somewhat simple method of identifying that something is happening, but not necessarily where it is happening—and may lead to the requirement for a broad structure inspection, which can be time-consuming. There have been complex methods of locating cracking in the past; these tend to use single-axis accelerometers mounted on multiple surfaces [34]. This adds additional complication as one has to calculate not just the crack location, but the 3D relative locations of one accelerometer to another.

Piezoelectric accelerometers are electromechanical transducers designed for measuring acceleration, shock, or vibration. When the accelerometer is subjected to vibrations, it triggers the inertial mass to ‘squeeze’ the piezoelectric materials, which produces an electrical current, which is proportional to the pressure applied to the material [51]. Their sensitivity is relatively low; hence they are suitable for sensing high frequency [52]. They have been used in numerous applications including earthquake detection [53], aerospace [54], medicine [55], and structural monitoring [56]. Piezoelectric accelerometers usually have a single component and measure acceleration in only one direction. Due to the recent advances in sensor technology, various new and improved sensors, such as triaxial accelerometers, have been developed for structural health monitoring of concrete structures [29,52]. Triaxial accelerometers combine three orthogonally orientated single-axis accelerometers into a single package and measure the full elastic acceleration wavefield in three orthogonal directions, usually termed as X, Y, and Z component directions [57]. Some of their advantages are that they are economical, light, are not affected by electromagnetic interference, and can be easily installed at a larger scale [52].

The aim of this study is to examine and evaluate the fracture behaviour of steel-fibre-reinforced self-compacting concrete (SFRSCC) beams under three-point bending using triaxial accelerometers. Two different types of beams were prepared, namely the plain SCC without fibres and SFRSCC beams. Triaxial accelerometers were used to collect data in real time for characterising their fracture modes. The aim is to examine the feasibility of using triaxial accelerometers for detection and localisation of cracks, and to provide an improved methodology for concrete structure condition assessment.

## 2. Experimental Materials and Methods

### 2.1. Material Properties

Ordinary Portland cement (CEM I 52.5N), conforming to British Standard (BS) EN 197-1:2011 [58] was used as the binder in the study. Crushed limestones with a maximum diameter of 14 mm were used for coarse aggregate. Natural sand and crushed sand with maximum diameter of 4 mm were used for fine aggregate. The steel fibres used in this study were the hooked-end type and prepared from cold-drawn steel wires according to BS EN 14889-1 [59]. Steel fibres were manufactured by Bekaert, with a length of 35 mm, a diameter of 0.75 mm, and at a dosage of 5 kg/m^3^ by weight of the cement. To achieve the desired workability in all concrete mixtures, a polycarboxylic ether polymer superplasticiser (SP) according to BS EN 934-2:2009 [60] was used. Two different SCC mixes were prepared, namely the plain SCC without fibres and SFRSCC. The mix proportions are shown in Table 1. To evaluate the workability of the SCC, slump flow, J-ring, and V-funnel tests were conducted as per EFNARC recommendations [61,62]. The mechanical properties were determined by curing the specimens for 28 days. Compressive strength was obtained according to BS EN 12390-3:2019 [63]. For the tensile strength, the tensile splitting test was performed as per EN 12390-6:2009 [64]. For each concrete mix, three cubic specimens of 100 × 100 × 100 mm were tested in each test. The fresh and mechanical properties of SCC and SFRSCC specimens are presented in Table 2 and Table 3, respectively.

### 2.2. Specimen Preparation

The test specimens consisted of both plain SCC and SFRSCC beams with dimensions of 500 × 100 × 100 mm. In total, six beams were tested, three of which were plain SCC and three were SFRSCC. The specimens were cast by pouring the freshly mixed concrete without compaction and kept in moulds for approximately 24 h. After demoulding, they were cured in water at a temperature of 20 ± 2 °C for 28 days prior to testing. Each time before testing, the test specimens were wiped with a damp cloth to remove excess water from the surfaces and air-dried at a room temperature of 20 ± 2 °C for approximately 30 min.

### 2.3. Experimental Setup and Testing Procedure

The three-point bending tests for the determination of the specimen flexural strength were conducted using a universal testing machine as per BS EN 12390-5:2019 [65]. The specimens were tested up to final failure by controlling the vertical displacement of the hydraulic jack with a rate of 2.2 mm/min. A general view of the experimental setup and the testing wiring diagram are presented in Figure 1 and Figure 2, respectively.

For the evaluation of natural frequencies for increasing loads and during crack propagation, the specimens were equipped with two triaxial piezoelectric accelerometers (AT/04/TB, DJB Instruments), positioned on the top surface of the beam at a 60 mm interval from either side of the mid-span. They were secured by the use of a thin layer of tacky adhesive during each test. The main characteristics of the adopted sensors included dimensions of 16.5 × 16.5 × 15.3 mm, frequency range of 1 Hz to 6 kHz, resonant frequencies of 33 kHz (z axis) and 20 kHz (x/y axis), operating temperature of −50 to +250 °C, and the specimen weight of 16.6 g. These accelerometers were used to identify and locate the process of crack formation by creating a three-dimensional vector in the form of orthogonal components. The data was collected by a PROSIG P8020 24-bit data acquisition device. The sampling frequency was set equal to 10 kHz. The background noise and interference were eliminated using a high-pass filter with a cut-off frequency of 40 Hz. This frequency was determined by trial-and-error and was found to be appropriate for the purpose of this study. All tests were conducted at an ambient temperature of 20 ± 2 °C and relative humidity of about 40%.

The method applied in this study is simple in its application: two triaxial accelerometers fitted to a surface of the beam. The only information required is the 2D location of the accelerometers relative to one another. This information can be as simple as ensuring they are on the same plane and knowing the distance between them on that plane. This specific data analysis was time-consuming and complex initially; however, one script with small edits could be used multiple times. There is also a large scope where with further work done by the research team this process could be broadly automated with the correct computer programming and even applied into a user-friendly graphic interface.

## 3. Results and Discussion

### 3.1. Mechanical Behaviour of Plain SCC and SFRSCC Beams

Figure 3 shows two different plots obtained from the three-point bending tests of plain SCC and SFRSCC beams. Figure 3a presents the force–time curves for the force measured by the loading cell at mid-span deflection, while Figure 3b shows the displacement-force curves for assessing the effect of fibre addition on the elasticity of SCC. As expected, the load-displacement curve is linear up to the maximum load for all specimens. As seen in Figure 4a, the behaviour of the plain SCC beam is typical for a brittle material with a linear force–time curve. The beam fails in tension with the crack initiating from the bottom surface and propagating towards the top, splitting it into two parts. Once the maximum load is achieved, any additional load will lead to the formation of the macrocrack and suffer a catastrophic failure [23,50].

The main observation of the differences on post-peak behaviour are as follows: (i) after reaching the maximum load, energy absorption in plain SCC beams is negligible; (ii) after achieving the peak load, there is a sudden load drop on SFRSCC beams; however, it continues to absorb energy to the end of the test while the load gradually decreases. An increase in the force can be observed in the force–time curve for the SFRSCC beam after the macrocrack has formed approximately 50 s into the test. This was due to the presence of fibres over the macrocrack region. Figure 4b shows the failure of SFRSCC beam where it did not break into two parts after final failure due to the presence of steel fibres. It is also worth mentioning that there is a succinct step down in the force–time curve due to fibres bridging the crack being gradually pulled out as the load continued to be applied. Similar results on the modes of failure of plain SCC and SFRSCC is also reported in Soulioti et al. [32].

As can be seen in Table 2, the SFRSCC beams exhibit a higher average flexural strength and a bigger deflection at maximum load than the plain SCC beams. This may be due to the increasing width of the fracture process zone (FPZ) caused by the distribution of fracture energy in a larger volume of material [32]. In addition, the SFRSCC beams have a higher elasticity than the plain SCC beams due to the inclusion of steel fibres.

### 3.2. Crack Detection and Localisation

In this section, filtered response data from three plain SCC beams and three SFRSCC beams were analysed. Further in-depth analysis of the results was then manipulated for a single plain SCC and SFRSCC in a variety of methods. The chosen beams have typical results to their equivalents. It is also worth noting that X1 and X2 in Figure 5, Figure 6, Figure 9, Figure 10, and Figure 11, represent the responses obtained from accelerometers 1 and 2, respectively. The benefit of using two triaxial accelerometers is that the specific crack location can be confirmed. The point of intersection, where the released energy from the source of the crack, captured by the two accelerometers, can be determined by a simple trigonometry, is discussed in Section 3.2.3.

#### 3.2.1. Amplitude–Time Curves

Figure 5a shows the amplitude–time curves obtained from the plain SCC beam with a clear notation showing the macrocrack. In addition, Figure 5b shows the microcrack at 32.5 s as a distinctive change in the behaviour during the test with a burst of energy shown by the increase in density of low amplitude peaks when compared to the surrounding data points. Figure 5c takes a closer look at the macrocrack, highlighting what was assumed to be a single distinct event to be at least four discrete events of varying duration, which is suggestive of multiple breaking or tearing events.

Figure 6a presents the amplitude–time curves from the SFRSCC beam. Figure 6b shows the microcracking with a similar behaviour to the microcracking in the plain SCC beam, with a series of low-amplitude peaks over a short period of time. This is then followed by a macrocrack denoted by two sections of large amplitude peaks. Figure 6c shows a fibre pull-out event which is easily characterised by its amplitude being larger than that of the microcrack, but smaller than that of the macrocrack, with no evidence of precursors.

In order to validate the technique used to determine the stages which define micro- and macrocracking the following steps have been made:The amplitude–time curves were aligned with the force–time curves, where the largest force peak time correlates to the largest amplitude peak time.The time at which the macrocrack visibly formed was documented in the testing notes and the time also correlated with the test results. The times were aligned therefore validating the macrocrack time.Once the time was validated, this was compared with the other stages of the AE trace.

In addition, the time–amplitude curves could also be used with the time to determine the crack type. Microcracking has a relatively small amplitude, which occurs prior to the macrocrack and has no visible precursors. The macrocrack had the clear precursors of microcracking; notably the fibre pull-out events had no detectable precursors and occurred after the macrocrack.

#### 3.2.2. Root Mean Square (RMS) Curves

The RMS curve is a measure of the overall signal energy used to extract key features of the signal and data trends. Figure 7a shows the RMS curves highlighting the macrocrack regions for the plain SCC beam, while Figure 7b shows those of the SFRSCC beam. The RMS curves clearly indicate that the amount of time the macrocrack takes to fully form reduces when steel fibres were added. Therefore, the higher percentages could withstand a lower maximum load and catastrophic failure occurs at an increased pace. The RMS curves shown in Figure 7a,b also identify the length of the macrocrack duration considered for further analysis. These curves start at the initiation of the macrocrack’s large energy burst, ending prior to any reciprocating waves. Notably, it shows a clear reduction in macrocrack duration when steel fibres were added.

#### 3.2.3. Hodograms

Hodograms have been used as a method to identify crack locations for plain SCC and SFRSCC beams. In addition to the requirement to plot each stage of cracking identified by the amplitude time plots, the locations of the accelerometers will cause each to receive the acoustic emission at a different time, therefore adding to the analysis complexity. Furthermore, there is only a small window to correctly plot the hodogram, and at points it can be as small as 0.0002–0.001 s in duration. The method for finding the correct time window is to visually identify a larger time window from the amplitude–time plots, then slowly reduce its size until the hodogram shows the required rectilinear notation which points directly towards the location of the crack. The hodograms have been used to derive the angle from which AE occurs with respect to the accelerometers as identified in Figure 8. The accelerometers were mounted at the distance of 60 mm from either side of the load point. The vertical distance from the load point was determined by using simple trigonometry. Figure 9 and Figure 10 shows the hodograms for the micro- and macrocrack stages for the plain SCC beam, while Figure 11 shows the results from the SFRSCC beam. All the beams from each concrete type showed similar characteristics and definable stages. Further experiments may provide more understanding of these characteristics. It is clear in the data that multiple clear stages of cracking occur. It would not be unreasonable to investigate the definitions of these stages more clearly with further research. Another experiment would be expected to show similar behaviour and definable stages.

Figure 12a illustrates the crack location trace for the plain SCC beam and it was created using the values from the hodograms. This trace shows each stage of the crack and where it is located on the beam with the bottom left-hand corner of the beam being the plot origin and the crack occurring at its mid-point (25 cm). The trace clearly shows the initiation of the crack at its lower surface and sequentially moves towards the upper surface where the load is applied. Notably, on the trace there is some areas where the cracking stages are mixed—this is due to errors in the margin or identification of the crack time. Irrespective of these errors, the trace still follows the expected trends of the crack emanating from the bottom of the beam (the area under tension) to the top of the beam (the area under compression) as concrete’s material properties lend better to compression rather than tension [62]. Figure 12b displays the crack location trace for the SFRSCC beam. This beam shows a much clearer trend to the plain SCC beam with defined areas of micro- and macrocracking occurring from the lower surface of the beam to the upper surface. The same markers are used to define the crack stages as used in Figure 12a to ensure the results are easily comparable. Furthermore, a physical representation of the hodogram plots for the SFRSCC beam is shown in Figure 13. The stages of cracking are indicated where they are physically occurring on the beam.

The hodograms use the initial part of the waveform, which presents as a rectilinear component and ‘points’ to the crack source. The later parts of the waveform present as circularly/elliptically polarised components and represent resonant modes of the clamped block. This is supported by the frequencies occurring as multiples of the beam’s dimensions. Furthermore, the order of the tear is supported by the hodogram analysis (bottom to top). However, some of the crack locations show values that lie outside of the beam’s boundaries. This is suggestive of a local scattering effect and could be a large piece of aggregate that causes rebounding waves, taking a different path to the original AE, thereby skewing the hodogram plot as shown in Figure 14. It must be acknowledged that concrete’s material properties are inhomogeneous in nature.

## 4. Conclusions

In the present study, triaxial accelerometers were used to detect and locate the fracture process in plain SCC and SFRSCC beams. The experiments involved plain SCC and SFRSCC beams tested under three-point bending and loaded until failure. Two triaxial accelerometers were used during the test to detect the natural frequency function of each beam. In terms of mechanical behaviour, SFRSCC exhibited higher strengths and improved post-peak behaviour compared to plain SCC. Waveform analysis was performed to determine the crack type, while hodograms were used to identify crack location. The experimental results demonstrated that the recorded responses from triaxial accelerometers correspond to the crack formation and propagation This technology may be suitable for broad applications; it would require further analysis and testing to clarify the applicability. Its use on a bridge deck would be a good example of where further research would be appropriate. As part of this investigation, the specific frequencies of the concrete cracks were detected. When installing a system like this on a road deck, the frequencies of interest would be programmed in, and as such, irrelevant background noise, i.e., from road traffic, would be removed from the system using appropriate filter techniques. When looking at other uses, the specific concrete cracking frequency could be added into the system and all other frequencies filtered out. This would remove the background noise from the system and allow for broader uses. This technique was also considered as part of this investigation and filtering was applied to assist in the removal of background noise from other equipment in the laboratory. This non-destructive method using triaxial accelerometers alongside waveform analysis and hodograms has led to improved accuracy in detecting and locating crack growth and propagation, and would provide a solution for a structural health monitoring system for concrete structures with easier applications, requiring less technical knowledge.

## Figures and Tables

**Figure 1 sensors-21-02044-f001:**
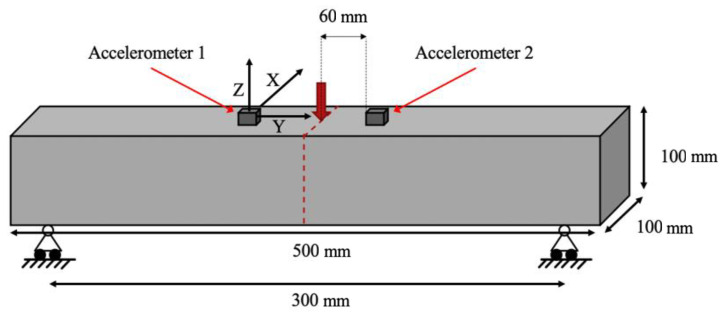
Experimental setup.

**Figure 2 sensors-21-02044-f002:**
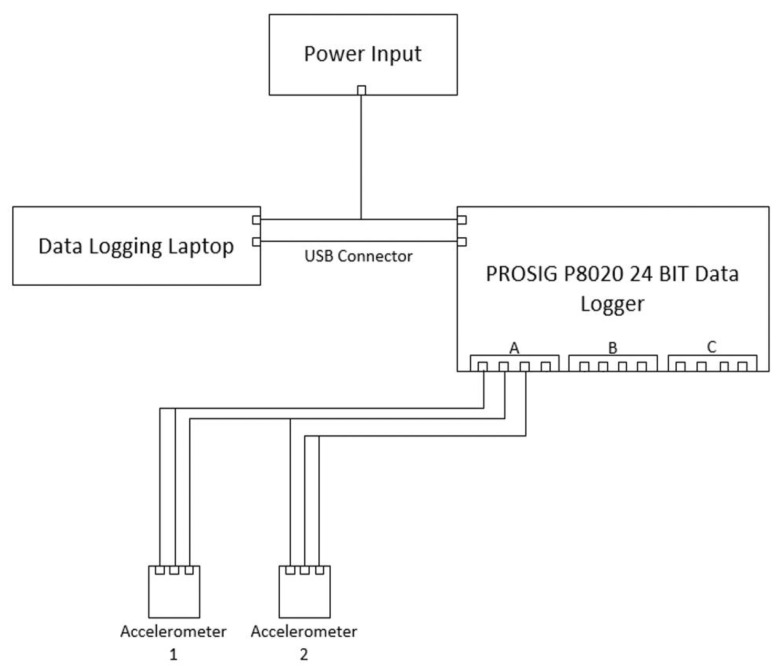
Testing wiring diagram.

**Figure 3 sensors-21-02044-f003:**
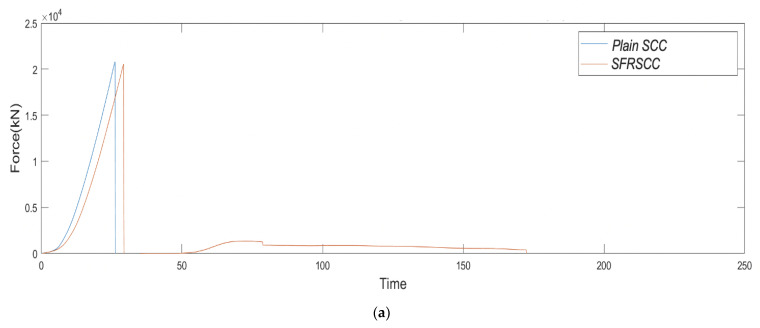
Three-point bending test results for plain SCC and SFRSCC specimens; (**a**) force vs. time curves, and (**b**) displacement vs. force curves.

**Figure 4 sensors-21-02044-f004:**
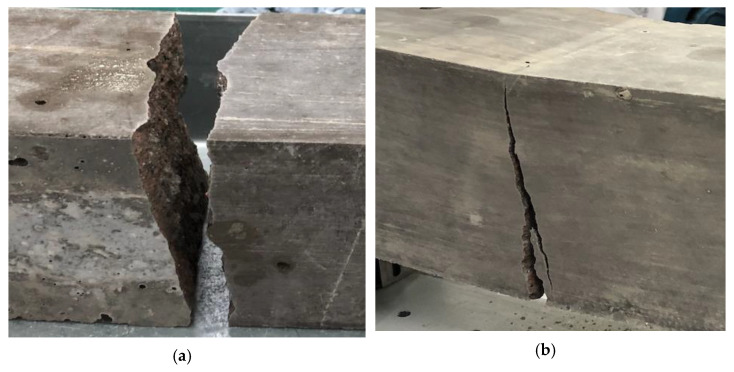
Modes of failure for: (**a**) plain SCC beam, (**b**) SFRSCC beam.

**Figure 5 sensors-21-02044-f005:**
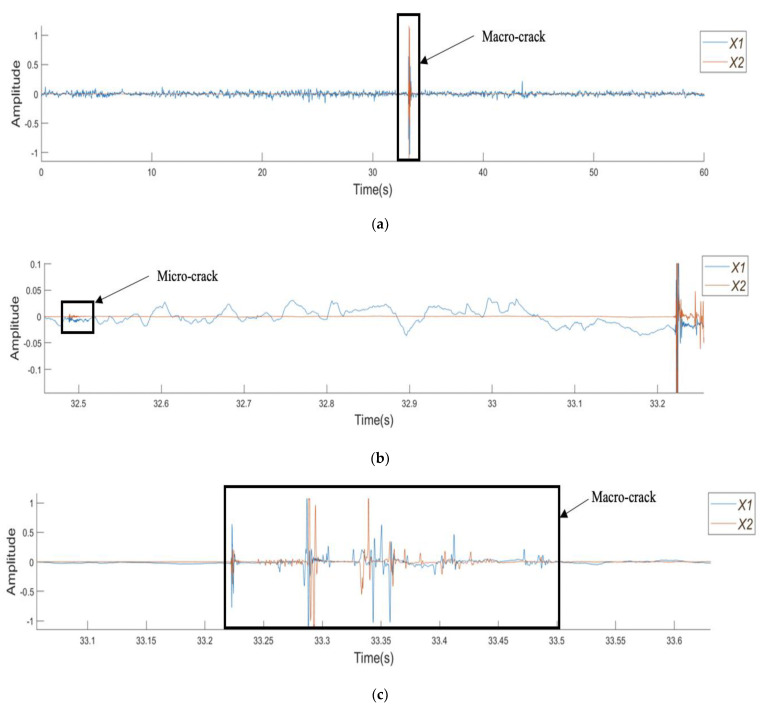
Amplitude vs. time plots for the plain SCC beam: (**a**) the complete test, (**b**) the section highlighting the microcrack, and (**c**) the section highlighting the macrocrack.

**Figure 6 sensors-21-02044-f006:**
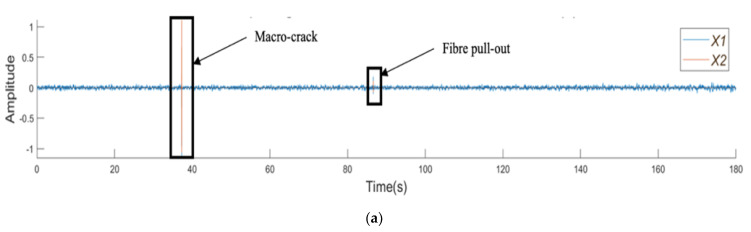
Amplitude vs. time plots for SFRSCC beam: (**a**) the complete test; (**b**) the section highlighting micro- and macrocrack regions; and (**c**) the section highlighting the fibre pull-out region.

**Figure 7 sensors-21-02044-f007:**
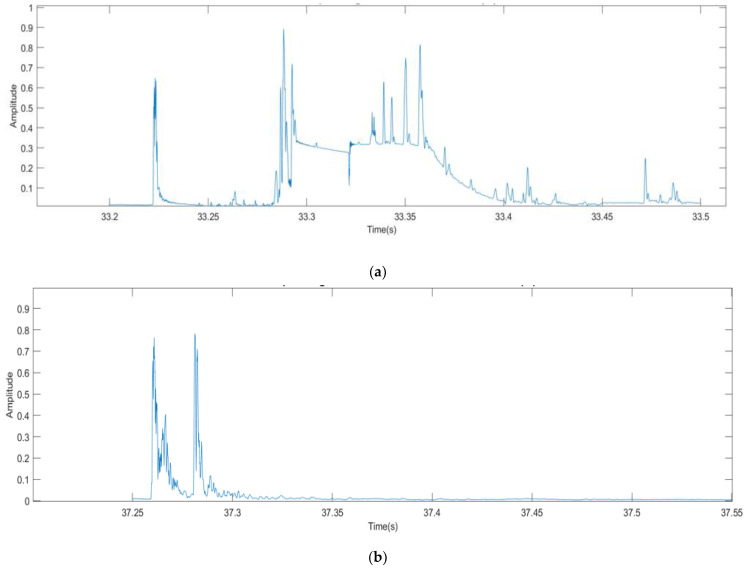
Root mean square (RMS) curves showing the range bars during the occurrence of macrocracks for: (**a**) plain SCC; and (**b**) SFRSCC beams.

**Figure 8 sensors-21-02044-f008:**
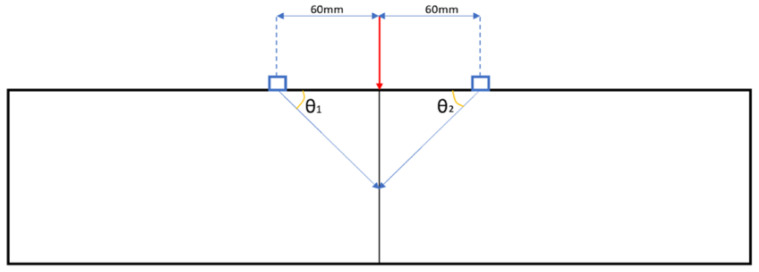
Hodogram crack location method.

**Figure 9 sensors-21-02044-f009:**
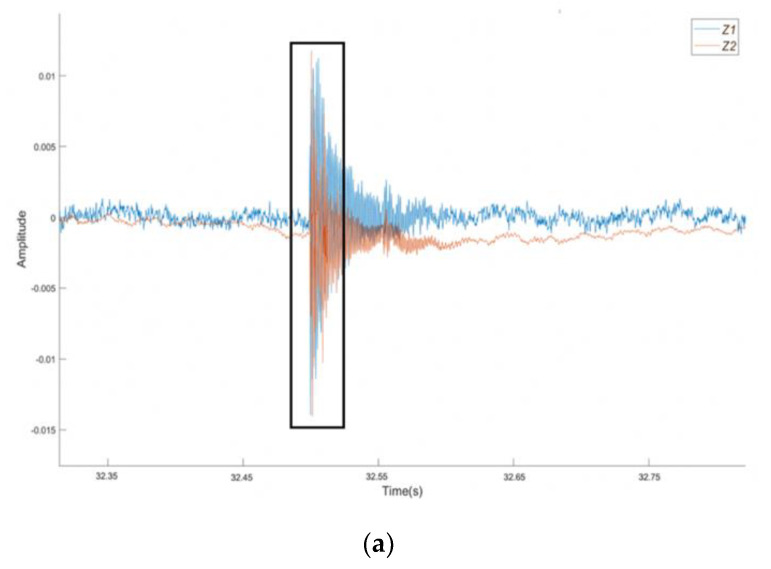
Hodogram analysis for the microcrack of the plain SCC beam: (**a**) amplitude–time curves highlighting the analysed section, and (**b**) hodograms for the microcrack.

**Figure 10 sensors-21-02044-f010:**
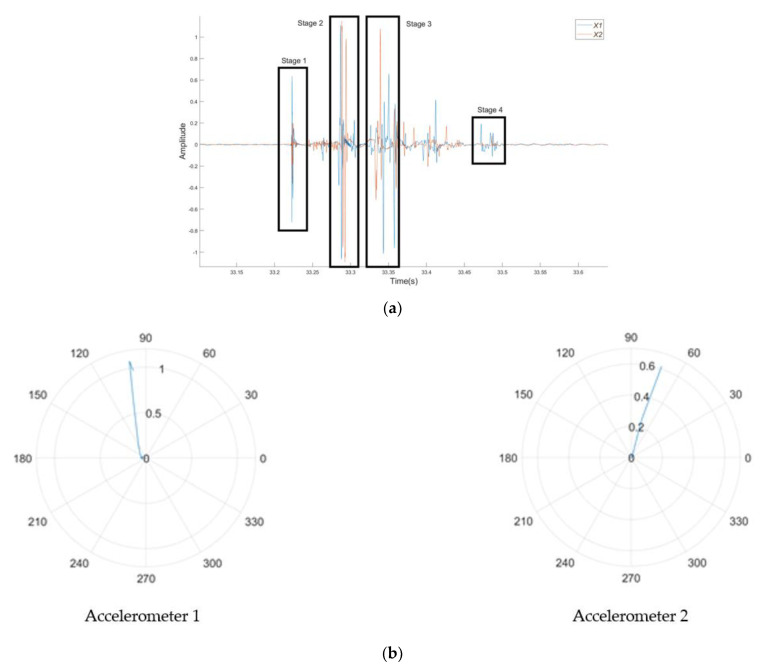
Hodogram analysis for various stages of macrocracks of the plain SCC beam: (**a**) amplitude–time curves highlighting the analysed sections; (**b**) hodograms for Stage 1 macrocrack; (**c**) hodograms for Stage 2 macrocrack; (**d**) hodograms for Stage 3 macrocrack; and (**e**) hodograms for Stage 4 macrocrack.

**Figure 11 sensors-21-02044-f011:**
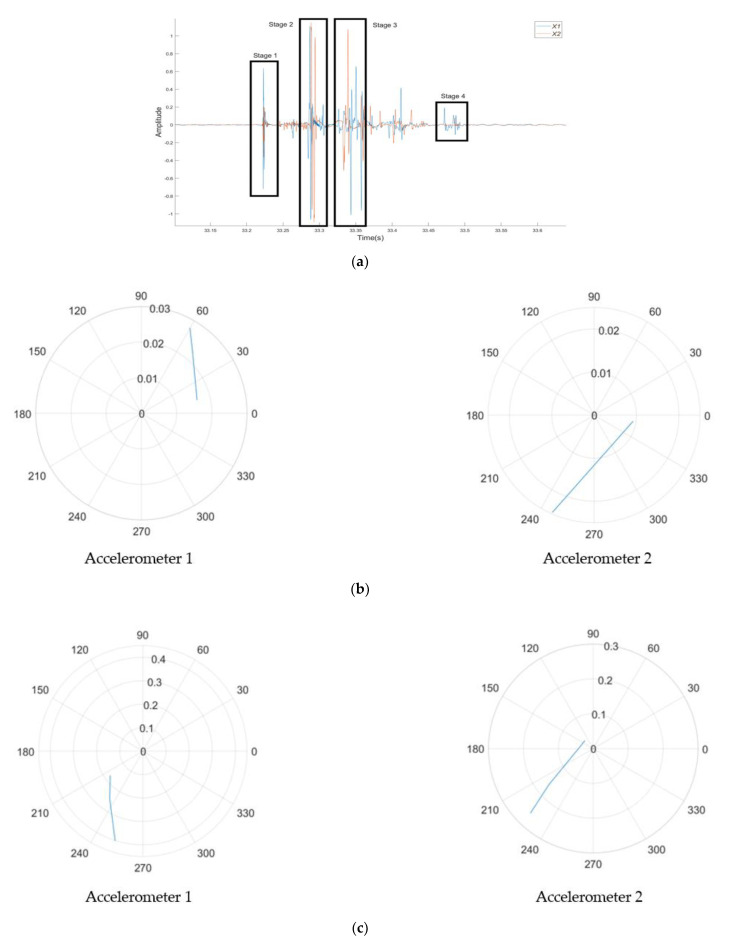
Hodogram analysis for the micro- and macrocrack of the SFRSCC beam: (**a**) amplitude–time curves highlighting the analysed sections; (**b**) hodograms for the microcrack, (**c**) hodograms for Stage 1 macrocrack, and (**d**) hodograms for Stage 2 macrocrack.

**Figure 12 sensors-21-02044-f012:**
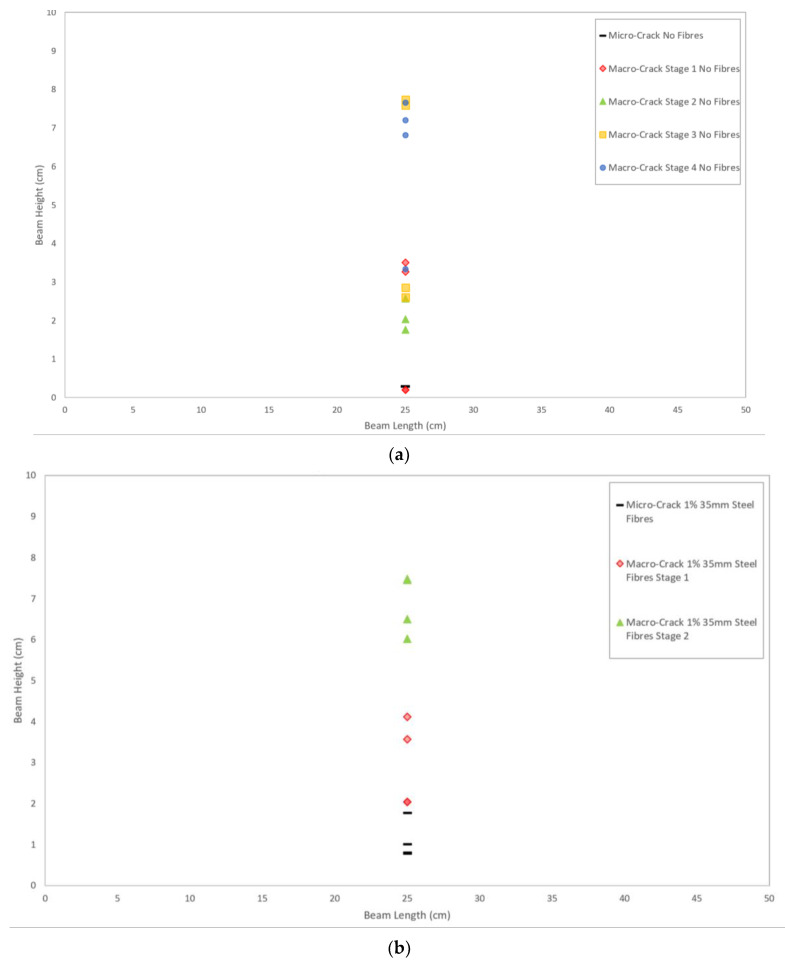
Crack location trace for: (**a**) plain SCC, and (**b**) SFRSCC beams.

**Figure 13 sensors-21-02044-f013:**
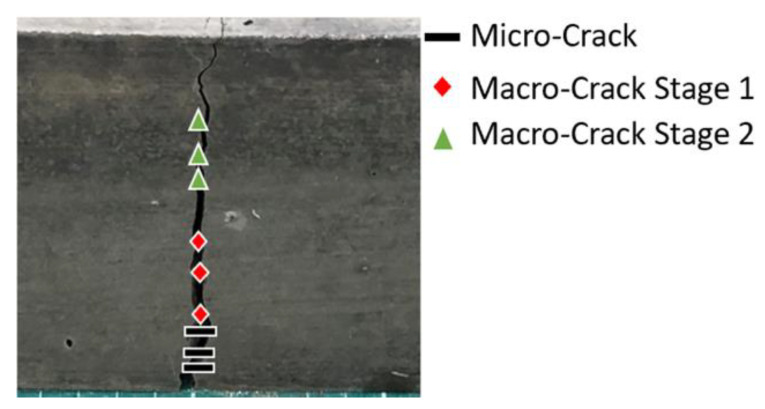
Physical representation of crack location trace for the SFRSCC beam.

**Figure 14 sensors-21-02044-f014:**
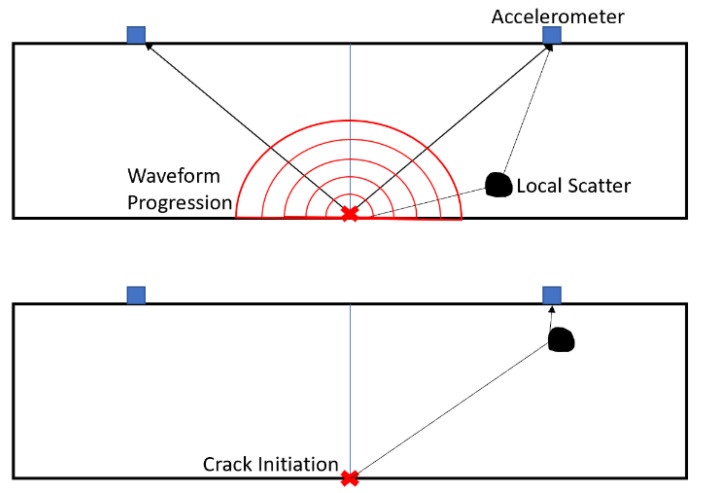
Local scatter diagram.

**Table 1 sensors-21-02044-t001:** Mix proportions of plain self-compacting concrete (SCC) and steel-fibre-reinforced SCC (SFRSCC).

Mix ID	Cement (kg/m^3^)	Limestone (kg/m^3^)	Natural Sand (kg/m^3^)	Crushed Sand (kg/m^3^)	Water (kg/m^3^)	Superplasticiser (kg/m^3^)	Fibres (kg/m^3^)
Plain SCC	480	795	668	265	187	3.95	0
SFRSCC	480	795	668	265	187	3.95	5

**Table 2 sensors-21-02044-t002:** Fresh properties for plain SCC and SFRSCC.

Mix ID	Slump Flow		V-Funnel Time (s)
	*D* (mm)	*t_500_* (s)	
Plain SCC	715	2.8	11.7
SFRSCC	650	4.6	20.1

**Table 3 sensors-21-02044-t003:** Mechanical properties for plain SCC and SFRSCC.

Mix ID	Compressive Strength (MPa)	Tensile Strength (MPa)	Flexural Strength (MPa)	Max. Load Deflection (mm)
Plain SCC	82.79	5.31	9.27	1.17
SFRSCC	92.84	5.43	9.61	1.35

## Data Availability

Not applicable.

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
