# Peer review of "Crack Detection and Localisation in Steel-Fibre-Reinforced Self-Compacting Concrete Using Triaxial Accelerometers"

_sensors, 2021, doi:10.3390/s21062044_

Round 1

Reviewer 1 Report

Thank you for your manuscript submission. This work shows a characterization method that characterizes concrete’s cracking behavior precisely and shows promise. I offer the following comments:

-In page 1, line 35, replace “On the contrary” with “However,” since the phrase “on the contrary” seems to suggest there is a contradiction with the literature with respect to the disadvantages of using fibers in SCC.

-In page 2, line 68, please spell out what AE means on the first time you introduce this term

-Are the steel fibers used in this study conforming to an ASTM or any other standardized design?

-How implementable is this technology? Would it be suitable for bridge deck monitoring, or is the instrumentation too sensitive for detecting cracks while the bridge deck is under vibrations from traffic loading? What are the scenarios in which this technology is suitable in real structures? Please provide more information on this matter in the manuscript.

-How does this proposed technology compare to other traditionally used devices to monitor cracking in terms of accuracy and ease of use?

-The manuscript concludes by mentioning that this proposed technology is a cost-comparable solution for SHM. How does this compare in cost with other traditionally used devices? There seems to be no mention of this elsewhere.  

Reviewer 2 Report

The manuscript presents a laboratory-based study to examine the efficiency of triaxial accelerometers in characterising fracture modes of beams made of self-compacting concrete (with and without steel fibres). 

The topic is interesting and the paper is well written. However, I would suggest the followings for further improvement:

  • explain the benefit/rationale for using 2 triaxial accelerometers.
  • type of steel materials used
  • Figure 6: does not clearly show the accelerometers which are supposed to be positioned on the top surface of the beam.
  • Figure 7: explain to the readers what are X1 and X2.

Figures quality are generally good but some of them should be improved on the text size (e.g. Fig 11 the numbers are too small to read) and some of them seem to be distorted (e.g. Fig 5).

Reviewer 3 Report

Dear Authors,

I have read your paper with pleasure and attention.

In my opinion, the manuscript Crack detection and localisation in steel fibre reinforced self-compacting concrete using triaxial accelerometers presents original research and innovative solution and could be interesting for readers of the MDPI Sensors Journal.

The paper presents an interesting approach in which triaxial accelerometers have been employed to detect and locate micro-and
macrocrack formation in plain self-compacting concrete and steel fibre reinforced self-compacting concrete beams under three-point bending. For this purpose, they have been used triaxial accelerometers mounted on the surface of the beams were used.

The motivation is clear. The object of study, as well as the results, are comprehensively described providing valuable conclusions.

I have no objections to publishing this paper. However, due to the listed below drawbacks, my recommendation is " Accept minor revision". In my opinion, several aspects require clarification. Please consider a minor revision and add some comments according to the following:

  1. The experiment was carried out for one plain SCC and one SFRSCC bar.  What is the repeatability of the measurement results? Can you comment on whether, according to the authors, in another experiment, it will be possible to distinguish stages 1, 2, 3, 4, 5 (Fig.13)?
  2. How sample humidity and ambient temperature may affect crack detection and localization accuracy.
